# Variations in anxiety and emotional support among first-year college students across different learning modes (distance and face-to-face) during COVID-19

**Flor Rocío Ramírez-Martínez[1], Maria Theresa Villanos[2], Sonam Sharma[2], Marie Leiner [1]***

**1** Department of Extension and Students Services. Universidad Autónoma de Ciudad Juárez, Chihuahua, México, **2** Department of Pediatrics, Texas Tech University Health Sciences Center, El Paso, Texas, United States of America

* marie.leiner@ttuhsc.edu

**Data Availability Statement:** The data will be available in a public repository database. The DOI/ accession codes will be made available at

## Abstract

College students with more emotional support experience lower levels of anxiety and other psychosocial and behavioral problems. During the COVID-19 pandemic, the emotional well-being of college students was additionally challenged by an abrupt shift to distance learning followed by a return to face-to-face classes. In this exploratory study, we compared the levels of perceived emotional support and anxiety among incoming first-year undergraduate students, prior to starting classes, which included different learning modes in five semesters of instruction from 2021–2023 (three distance semesters and two face-to-face semesters). Data from 8659 undergraduate students were extracted from a Mexican state university database, corresponding to students' responses collected during new student orientation week. Participants were students in the arts and humanities (9.7%), social and legal sciences (38%), life and health sciences (28.9%), and engineering and architecture (23.4%). Anxiety levels were measured with the GAD-7 scale, and emotional support was measured using a subscale of the PERACT-R (To go through with resilience) inventory. Comparisons of emotional support and anxiety scores among semesters revealed highly significant differences with small effect sizes. Anxiety levels increased significantly with mean average of 6.65 SD(5.52) during the baseline measure to the highest in 2022–2 to 7.53 SD(5.3) and Emotional Support decreased systematically each semester from baseline mean = 8.03 SD(2.0) to the lowest 7.52 SD(1.8) in 2022–2. The results show that a return to face-to-face classes was associated with increased anxiety levels, whereas levels of emotional support systematically decreased across the five semesters. MANOVA analysis revealed significant differences in anxiety and emotional support scores between semesters, with peaks during the learning mode semester that students returned to face-to-face classes after distance learning even after adjusting for gender. Given that the effect of emotional support on anxiety may be related to success in future educational and professional activities, it is important to develop interventions to restore and increase college students' emotional support levels and develop anxiety management strategies.

acceptance. Data will be available from the Zenodo database [Data set]. Zenodo. https://doi.org/10. 5281/zenodo.8250874.

**Funding:** The author(s) received no specific funding for this work.

**Competing interests:** NO authors have competing interests.

## Introduction

Studies have shown that college is a challenging time for students, with lasting effects on their quality of life and mental health, influenced by factors such as academic discipline, personal and/or contextual factors [1, 2]. Students in health sciences disciplines have traditionally been considered to face different challenges than their counterparts in non-health disciplines regardless of geographical location, as a result of the academic demands, time commitment and rigorous exams, as well as the pressure exerted by the clinical education environment [2–6]. Although academic disciplines may present different challenges, evidence suggests that students, regardless of field of study and/or location, can experience high stress, anxiety, depression, and low quality of life, potentially affecting their academic and social functioning [7, 8]. Starting college can involve significant social changes in the lives of students; for example, they stop being teenagers and become adults, acquiring a certain independence, which is often accompanied by new demands and responsibilities [9–11].

Pressures to excel academically, make new friends, and establish a sense of identity can contribute to college students' anxiety [12, 13]. In addition, the stress of living away from home and dealing with finances can further exacerbate anxiety [14]. Research has shown that anxiety is prevalent amongst college students but it can be further heightened by contextual factors in times of uncertainty, such as during the COVID-19 Pandemic [15]. COVID-19 significantly disrupted teaching and learning practices, as college students were forced to quickly adapt to a socially isolated distance learning environment [16, 17]. This shift to online learning modes led to a decline in academic performance as students lost the structure and support of a traditional classroom environment [18]. In addition, social distancing limited opportunities for in-person interaction, making it difficult for students to establish new friendships and experience a sense of community [19–21]. Due to school closures, students also lost crucial access to support systems, such as socializing with peers, which further intensified their mental health problems [22–24].

The adoption of distance learning during the COVID-19 pandemic resulted in poor academic performance and further increased anxiety among college students [25]. Distance learning proved difficult for students vulnerable to other online distractions, especially those with attention deficit hyperactivity disorder [23]. Distance learning widened the digital divide, creating new challenges for students who lacked access to reliable technology and the Internet, thus limiting the quality of the learning experience. Moreover, one study reports that in disadvantaged communities, only 42% of teachers had the necessary training or knowledge to use software programs to provide adequate instruction to students [18]. This lack of structure and accountability in e-learning resulted in decreased motivation, commitment, and attendance [26]. Consequently, math and reading test scores for students at all grade levels declined, with a widening of disparities based on students' racial, ethnic, and socioeconomic backgrounds [24, 27, 28].

When the initial wave of the COVID-19 pandemic subsided and universities reopened for face-to-face classes, many college students faced significant challenges and increased anxiety [29–31]. The sudden transition from distance learning to face-to-face classes, combined with fear of contracting the virus, contributed to increased stress levels [32]. In addition, students felt unprepared for the academic rigors of face-to-face classes after a long period of distance learning mode. Adapting to new health and safety protocols, such as the use of face masks and social distancing, also contributed to the challenges [33]. In addition, students had to cope with the risk of exposure to the virus, especially if they lived on campus or attended face-to-face classes [34, 35]. These challenges created a sense of uncertainty and discomfort, making it difficult for university students to focus on their studies and maintain their mental health.

## Review of related research

Emotional support has been considered crucial for helping college students during challenging times and reducing their anxiety [36–40]. Theory and research findings indicate the importance of satisfying relationships with peers and family in predicting the academic performance and persistence of college students [41]. The theories we are referencing here frame sense of support as the most active ingredient of social support. This argument is based on individuals' belief that they have people who value and care about them and are willing to try to help them. Although support from others differs in its ability to communicate or provide love, knowing that one is loved and that others will go out of their way to help us and when problems arise may be the essence of social support [42]. Social and emotional support appears to mediate somatic anxiety symptoms experienced by first-generation students [43]. Previous studies show that students who participate in peer support programs emphasizing emotional and social support have lower levels of anxiety, depression, and stress compared with students who do not participate [44]. Students who report lacking psychological support from their families also show a higher likelihood of anxiety compared with other students [45].

Based on the evidence presented, it is likely that many students from all fields of study, living in different economic contexts, experienced higher levels of anxiety in the years surrounding the pandemic, and that variations occur due to the different modes of learning—distance and face-to-face. It is also possible that freshmen who either completed or did not complete the previous level and began college in a distance or face-to-face learning mode were also affected. In addition, it is possible that the emotional support students felt, and levels of anxiety varied at different times during the pandemic and, more importantly, that systematic learning about students' struggles may enable the development of projects to ensure that students are offered appropriate services and interventions. This study analyzed data collected anonymously over several semesters from students starting college during new student orientation week prior to start classes in an upper-middle-income country during COVID. Specifically, we sought to observe the variations in anxiety and emotional support across different learning modes including distance and face-to-face, during COVID-19 among first-year college students. To our knowledge, there is insufficient information on the relationship between perceived emotional support and anxiety levels among these students during COVID collected over multiple semesters of admission when classes were offered in a distance or face-to-face learning mode. We hypothesized that anxiety would be negatively associated with emotional support among incoming college students, regardless of whether they started attending distance or face-to-face classes. In addition, anxiety levels would be higher for incoming students upon transitioning back to face-to-face classes, and emotional support levels would be perceived to be similar for students who started college in either of the semesters considered.

Hypothesis:

H1: There would be a variation in anxiety levels for incoming starting college students when transitioning from distance to face-to-face instruction modes.

H2: There would be similar levels of perceived emotional support for incoming starting college students across semesters offered in a distance and face-to-face mode format.

## Materials and methods

### Ethical approval

Institutional Review Board (IRB) of Texas Tech University Health Sciences Center El Paso, Texas, USA, approval was requested for the protocol including data review and analysis. It was

concluded that this study did not meet the definition of a human research study as all data were anonymized and no individual could be identified, IRB approval was not necessary. No further review was required.

## Study design

This study analyzed data collected from a systematic, anonymous, voluntary survey of incoming college students who began their studies at the Universidad Autonoma de Ciudad Juarez, Mexico, in either the spring or fall semester in the largest campus (North). It included independent data collected over five semesters, spring and fall 2021, 2022, and spring 2023 from the north campus. Data were collected prior to the start of classes during new student orientation week, which includes several information sessions for students and/or parents to familiarize them with offered services, programs, etc. During the pandemic, these sessions were conducted in distance learning (3 sessions) or face-to-face learning modes (2 sessions) concurrently with their class delivery formats.

Located on the northern border with the United States in the state of Chihuahua, the Universidad Autonoma de Ciudad Juarez is a public teaching and research institution. It is the largest university in a city of about 1.5 million inhabitants. The university is spread over four campuses, two outside the city, and two in the city. The north campus is composed of 57 programs distributed in the schools of architecture, design, and art; biomedical sciences; social and administrative sciences; and engineering and technology. The faculty of all campuses is about 816 full-time faculty and 36,500 students with a percentage of women students at 56%.

Regarding the socioeconomic level of the students attending the university, it is estimated that most of the students have limited economic resources. In fact, according to a report by CONEVAL, the National Council for the Evaluation of Social Development Policy [46], the city's population remained below the extreme poverty line during all the semesters in which the study was conducted.

## University psychological support program and contextual factors

This institutional program proposed and supported this data collection. It is a program that provides psychological support to students who are experiencing emotional, family, or social difficulties. In addition, students can use a mobile application to request support and counseling securely and confidentially, as well as participate in distance or in-person conferences in areas of personal and academic improvement offered each term. This program, along with other services and information about the university in general, is announced to incoming students during new student orientation week prior to the start of classes. To better understand the needs of students, anonymous surveys were administered during the pandemic, requesting information about anxiety, resilience levels, and media use, with assurances that the information provided was confidential. The students were also informed that the purpose of collecting this confidential, voluntary, non-identifying information was to provide them with support programs, which were almost simultaneously launched including a student wellness program that has been in place for several semesters (results of the program not included).

Lockdown in Ciudad Juarez started around October 2020 due to red traffic light, a monitoring system for epidemiological risk of COVID-19. A curfew was set to take effect as part of restrictions imposed in Chihuahua state including Ciudad Juarez because the region was experiencing a major surge of Covid-19 infections [47]. The overnight curfew in Juárez was running from 10 p.m. to 6 a.m. every day, stopping people from holding parties, limiting capacity in restaurants, stores, churches, movie theaters, libraries, children parks, museums and cultural centers. Alcohol sales were prohibited from Thursday to Sunday in the hopes of

preventing gatherings. Face masks and social distancing was required although fines were not implemented. Border crossing between USA-Mexico was restricted for 19 months starting March 2020 [48]. Remote learning was extensive at the high school and college level and occurred simultaneously from March 2020 to August 2022.

## Participants

The original data (N = 8,659) were obtained from a university database in Mexico containing de-identified information from surveys administered to starting college university students for five semesters during the COVID-19 pandemic from Spring 2021 to Spring 2023. Data from five semesters were analyzed, which included three semesters of distant learning classes (Spring 2021, N = 1782; Fall 2021, N = 2019; Spring 2022, N = 1667) and two semesters of face-to-face classes (Fall 2022, N = 2287; Spring 2023, N = 904). Deidentified information from the survey responses were obtained from university students during the university's new student orientation week prior to classes and stored in a database. The authors of the article or university authorities did not have access to identifiers that could link the data provided back to the students who completed the information.

## Measures

Demographic information extracted from the database included gender, age, study area, occupation, living situation, semester, and class format (distant or face-to-face). The data stored in the database was accessed once the information for the spring 2023 semester was collected.

*Anxiety*. The General Anxiety Disorder Scale (GAD-7) [49] is a seven-item self-administered instrument that uses some Diagnostic and Statistical Manual of Mental Disorders, Fifth Edition criteria for general anxiety disorder to identify the probable occurrence and measure the severity of anxiety symptoms. GAD-7 scores of 5–9, 10–14, and 15–21 are considered to indicate mild, moderate, and severe levels of anxiety. The validity of the GAD-7 instrument is demonstrated by robust correlations with other domains of functional impairment. Cronbach's alpha indicates excellent internal consistency of the instrument (Cronbach's alpha = 0.90) [49].

*Emotional support*. Emotional support was measured by the emotional support subscale from the PERACT-R (To go through with resilience) resilience inventory. The full scale measures the extent to which individuals have developed key abilities to respond to and cope with difficult situations [50]. The emotional support subscale includes three items: 1) "When you are faced with a problem or a difficult situation, do you have one or more people to help you find solutions?", 2) "When faced with a problem or difficult situation, do you look to spirituality (meditation, religion, yoga, etc.) for the strength you need to solve/cope with those problems?", and 3) "When you are faced with a problem or a difficult situation, do you have one or more people to advise you on how to find a solution to your problems?". The items are measured on a 4-point Likert scale as 1 = never, 2 = sometimes, 3 = often, and 4 = always. The emotional support score was calculated from the sum of the three item scores, with higher scores indicating more emotional support. Based on a pilot test, the results of confirmatory factor analysis (CFA) indicated that this subscale had good model fit (root mean square root of the error of approximation (RMSEA) = .053, 90% confidence interval 0.052–0.055; comparative fit index (CFI) = 0.91; Tucker Lewis fit index (TLI) = 0.89; normed fit index = 0.91; goodness of fit = 0.96; adjusted goodness of fit = 0.95; standardized root mean residual = 0.04). Cronbach's alpha indicated excellent internal consistency of the subscale (Cronbach's alpha = 0.90).

## Procedure

Every semester recruitment (Spring 2021; Fall 2021; Spring 2022; Fall 2022; Spring 2023) occurred during new student orientation week at a university in Mexico located near the Mexico-US border. New students responded voluntarily and anonymously to several surveys lasting approximately 15 minutes as part of program activities. Responses to the GAD-7, PER-ACT-R emotional support subscale, and some additional demographic data were extracted for the present study. The surveys were administered to incoming freshman students over five consecutive semesters from 2021 to 2023, which included both learning modes distance and face-to-face semesters. Voluntary responses included in the database varied between 32 to 48% of the total of students over the five semesters.

## Statistical analysis

Data were analyzed using the IBM SPSS Statistics 24 program. Continuous data were reported using mean and standard deviation (SD), and categorical data were reported using frequency and proportion. The data were analyzed for the presence of multivariate atypical values as well as normal distribution. Examination of histograms, skewness, and kurtosis of variables revealed deviations from normal distributions. Anxiety and emotional support scores were log-transformed and compared among semesters using multivariate analysis of variance (MANOVA) adjusted for gender. As some non-normality existed after transformation, we used a threshold of $p = 0.008$ to indicate statistical significance. In addition, as the F tests for MANOVA were highly significant ($p < 0.001$), discriminant analysis was performed to further characterize the dimensions on which the semesters differed. The structural validity of the emotional support subscale was determined by performing CFA using AMOS Graphic (Version 24.0). The CFA model was estimated using maximum likelihood parameter estimates with standard errors and a mean-adjusted chi-square test statistic that is robust to non-normality. CFA models were evaluated for different fit indices: root mean square root of the residual (RMR), RMSEA, TLI, and CFI. Generally, fit is considered acceptable when RMR and RMSEA values are $< 0.08$ [51] and TLI and CFI values are $> 0.90$ [52]. Spearman's correlation coefficient, r, was used to assess the association between social support, anxiety, and distance or face-to-face classes over five semesters.

## Results

The mean age of participants below 25 years old was 92.7% with a range of 18–39 years old. The sample included 5198 (60.0%) females and 3461 (40.0%) males. Table 1 shows demographic characteristics of participants across semesters. Distributions with respect to age, gender, civil status, place of residency, living status and area of study were comparable across semesters, although some differences were found that are not unusual due to the fact that responses are voluntary. An average of 75 to 78% of the students are originally from Ciudad Juarez, most of them indicated to be single and the study area with the highest percentage of students was Social and Legal Sciences across semesters followed by Life and Health Sciences.

Table 2 includes the non-transformed anxiety and emotional support scores of participants across semesters to simplify, in addition to anxiety levels categorized as mild, moderate, and severe. Anxiety and emotional support scores differ significantly across semesters. Post hoc comparisons across semesters show that the Fall 2022 semester has the higher scores in terms of anxiety scores as well as the proportion of moderate and severe anxiety levels. A decrease in emotional support scores each semester is observed across semesters, with significant differences between Spring 2021 and all considered semesters.

**Table 1. Participant socio-demographic characteristics for each semester.**

|  | 2021 | | 2022 | | 2023 | |
| --- | --- | --- | --- | --- | --- | --- |
| Variables | Spring | Fall | Spring | Fall | Spring | |
|  | N = 1782 | N = 2019 | N = 1667 | N = 2287 | N = 904 | p-value |
| Gender, n (%) | | | | | | |
| Male | 687(38.6) | 764(37.8) | 686(41.2) | 954(41.7) | 370(41.9) | 0.049 |
| Female | 1095(61.4) | 1255(62.2) | 981(58.8) | 1333(58.3) | 534(59.1) | |
| Age (%) | | | | | | |
| < 25 years | 90.3 | 95.1 | 86.7 | 92.9 | 93.9 | < .001 |
| Civil status (%) | | | | | | |
| Single | 91.5 | 96.0 | 92.1 | 96.3 | 93.6 | < .001 |
| Residency (%) | | | | | | |
| Cd. Juarez | 77.3 | 76.4 | 78.2 | 74.8 | 78.2 | .086 |
| Living (%) | | | | | | |
| Alone | 2.0 | 0.9 | 1.4 | 1.0 | 1.3 | .007 |
| One parent | 16.1 | 13.9 | 13.9 | 12.4 | 13.7 | |
| Family/friends | 81.9 | 85.2 | 84.6 | 86.6 | 85.0 | |
| Study Area (%) | | | | | | |
| A & H[AH] | 8.4 | 8.6 | 7.7 | 10.8 | 15.8 | < .001 |
| S & L[SL] | 36.5 | 40.9 | 35.4 | 40.0 | 34.3 | |
| L & H[LH] | 35.0 | 25.3 | 33.8 | 23.1 | 30.2 | |
| E & A[EA] | 20.1 | 25.2 | 23.0 | 26.1 | 19.7 | < .001 |

AH = Arts and Humanities, SL = Social and Legal Sciences LH = Life and Health Sciences EA = Engineering and Arquitecture

To assess whether female or male students had different anxiety and emotional support scores over different semesters of distance and face-to-face learning modes during the pandemic and whether there was an interaction between gender and semesters, two-way MANOVA (multivariate analysis of variance) was run with semester and gender as independent variables and anxiety and emotional support non-transformed scores as dependent variables. The interaction effect between gender and semester on the combined dependent variables was not statistically significant (F(8, 17296) = 1.92, p = 0.052, Wilks' $\Lambda$ = 0.998, partial $\eta^2$ = 0.001). The main effect of gender on the combined dependent variables was significant (F(2, 8648) = 199.26, $p < 0.001$, Wilks' $\Lambda$ = 0.956, partial $\eta^2$ = 0.044), indicating that anxiety and emotional

**Table 2. Participants non-transformed anxiety and emotional support scores across semesters.**

|  | 2021 | | 2022 | | 2023 | |
| --- | --- | --- | --- | --- | --- | --- |
| Variables | Spring | Fall | Spring | Fall | Spring | |
|  | N = 1782 | N = 2019 | N = 1667 | N = 2287 | N = 904 | p-value |
| Anxiety Mean (SD) | 6.65(5.2) | 6.90(5.1) | 6.6(5.1) | 7.53(5.3) | 6.2(5.0) | <0.001* |
| Emotional Support Mean (SD) | 8.03(2.0) | 7.77(1.9) | 7.66(1.9) | 7.52(1.83) | 7.62(2.0) | <0.001* |
| Anxiety levels (%) | | | | | | |
| Mild | 30.5 | 32.1 | 32.0 | 33.5 | 31.0 | <0.001** |
| Moderate | 13.2 | 14.8 | 13.6 | 16.8 | 11.6 | |
| Severe | 8.1 | 7.7 | 7.3 | 9.8 | 7.1 | |

*Significant from ANOVA Test.

** Significant from Chi square analysis

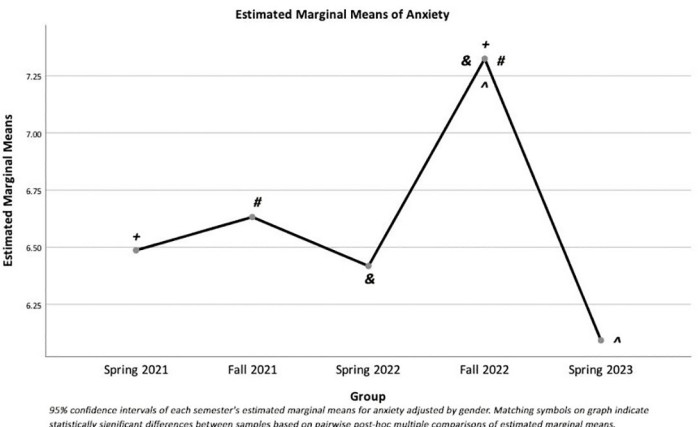

**Fig 1. Estimated marginal means for anxiety by semester adjusted by gender.**

non-transformed support scores differed between males and females. The main effect of semester on the combined dependent variables was also significant (F(8, 17296) = 14.61, p < 0.001, Wilks' Λ = 0.987, partial η2 = 0.007), indicating that the linear composite differed across semesters. Next, follow-up univariate two-way ANOVAs were run including the main effects of semester and gender.

**Hypothesis**:

H1: There would be a variation in anxiety levels for incoming starting college students when transitioning from distance to face-to-face instruction modes.

H2: There would be similar levels of perceived emotional support for incoming starting college students across semesters offered in a distance and face-to-face mode format.

There were significant main effects of semester on anxiety scores (F(4, 8649) = 13.65, p < 0.001, partial $\eta^2$ = 0.006) (Fig 1) and emotional support scores (F(4, 8649) = 17.30, p < 0.001, partial $\eta^2$ = 0.008) (Fig 2). Additionally, there were significant main effects of gender on anxiety scores (F(1, 8649) = 341.95, p < 0.001, partial $\eta^2$ = 0.038) and emotional support scores (F(1, 8649) = 23.03, p < 0.001, partial $\eta^2$ = 0.003).

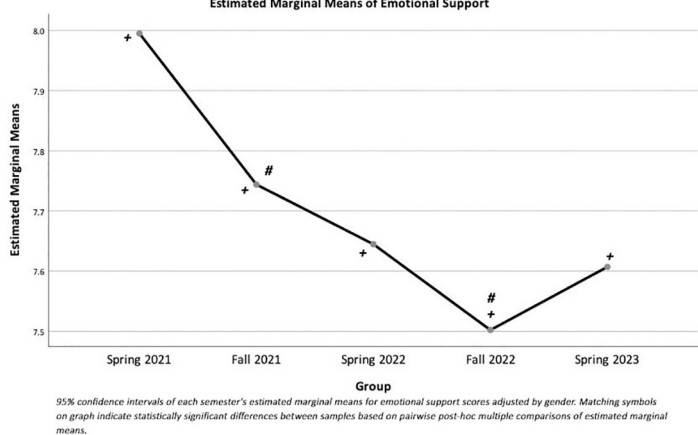

**Fig 2. Estimated marginal means for emotional support by semester adjusted by gender.**

MANOVA analysis showed that anxiety and emotional support scores were significantly different among semesters, even after adjusting for gender. One-way ANOVAs comparing individual raw scores showed significant differences in anxiety scores between semesters with distance vs. face-to-face classes. In particular, anxiety scores in the Fall 2022 semester, when students returned to face-to-face classes, were significantly higher than anxiety scores in all other semesters (Fig 1). Anxiety scores were also significantly higher in the distance Fall 2021 semester than in the Spring 2023 face-to-face semester. Emotional support scores decreased systematically across semesters, with significant differences between the initial measures across all the semesters (Fig 2).

### Anxiety and emotional relationships

Pearson's correlation was used to determine the relationships between anxiety and emotional support. Significant negative relationships were found per semester 2021 Spring ($r = -0.156$, $p < 0.001$) Fall ($r = -0.100$, $p < 0.001$) 2022 Spring ($r = -0.144$, $p < 0.001$) Fall ($r = -0.114$, $p < 0.001$) and 2023 Spring ($r = -0.139$ $p < 0.001$).

MANOVA was followed up with discriminant analysis, which revealed two discriminant functions. Both discriminant functions were significant (Wilks's $\Lambda = 0.986$, (8) = 125.06, $p < 0.001$ for discriminant function 1; Wilks's $\Lambda = 0.996$, (3) = 33.64, $p < 0.001$ for discriminant function 2). The first discriminant function explained 73.2% of the variance, and the second discriminant function explained 26.8% of the variance. Discriminant function 1 had the largest absolute correlation with emotional support, followed by anxiety. Discriminant function 2 had the largest absolute correlation with anxiety, followed by emotional support. Pairwise comparison of function 1 scores showed differences between the following semesters: Spring 2021 (M = 0.155) vs. Fall 2021 (M = 0.005), Spring 2021 (M = 0.155) vs. Spring 2022 (M = 0.013), Spring 2021 (M = 0.155) vs. Fall 2022 (M = -0.145), and Spring 2021 (M = 0.155) vs. Spring 2023 (M = 0.027). In addition, pairwise comparisons of function 2 scores showed differences between the following semesters: Spring 2021 (M = 0.055) vs. Fall 2021 (M = 0.030), Spring 2021 (M = 0.055) vs. Spring 2022 (M = -0.066), Spring 2021 (M = 0.055) vs. Fall 2022 (M = 0.32), and Spring 2021 (M = 0.055) vs. Spring 2023 (M = -0.137).

## Discussion

This study analyzed data from a large university database containing survey responses from new college students regarding their anxiety and emotional support prior to attending classes distance or upon return to face-to-face classes during the COVID-19 pandemic. Students' anxiety levels peaked in the first semester that classes returned to a face-to-face format. During the second semester of face-to-face classes, anxiety levels decreased. As in previous studies [53], anxiety levels of females were higher than those of males, with a consistent trend across all semesters.

Although all students at any educational level experience anxiety, it is important to note that the pandemic posed an additional challenge for students who transitioned between educational levels. In the study, some students completed distance learning in high school and entered college in the same format. Others went on to complete a distance-learning high school and begin college in a face-to-face program. Starting a different educational level is stressful because it requires a change in routine, teachers, friends, and financial conditions. Internal reports from the Psychological Support Program in the University indicated that levels of stress and anxiety were high among students in general. It seems that the Mexican students in this study appeared to have elevated levels of anxiety (score of > 10 on the GAD-7) compared with university students in China or Hong Kong in the same year (21.3% vs. 9.7%,

respectively) [54]. These high anxiety levels information suggest the need for the university to incorporate strategies including one that seemed to attain positive results that include a mandatory attendance to a Wellness Workshop offered to starting students in the first weeks of class to help reduce their anxiety and stress. The results of this study are not presented in this paper.

Regarding emotional support, there was a systematic decrease in scores across semesters, although the causes of this decline are unknown. Levels of emotional support in the semester that students returned to face-to-face classes showed a similar pattern as anxiety scores, although emotional support scores did not recover in later semesters to the same degree as anxiety scores. Consistent with previous studies [55], male students reported a lower emotional support scores than female students. The emotional support of students is crucial for favorable outcomes, such as academic and job performance, psychological health, and reduced suicide risk [56]. Evidence suggests that family support is very important for Mexican families, with studies indicating that this also holds true for families in the US [57].

The available evidence indicates that university students at the beginning of their careers or in later years worldwide, regardless of their field of study, are vulnerable to mental health problems and that contextual factors such as the pandemic have a significant influence. For this reason, universities implement strategies that provide students with the resources necessary to overcome these problems and achieve academic and personal success, despite the challenges involved [58]. Some universities provide counseling services to help students in need, which unfortunately are not being utilized as one would expect. A study conducted during COVID reported that the majority of students with moderate or severe mental health symptoms never utilized internal or external mental health services [59]. Another valuable strategy for universities includes the systematic collection of anonymous information from students regarding their quality of life, and mental health which can guide further strategies to address their problems. Anonymous information is more feasible to be collected while students know their names are not disclosed and is an opportunity also to offer during this data collection the counseling services they can access [60]. Resulting information allows for general and specific programs that can be offered to the students and should be always available for the students.

Some universities are capitalizing on the fact that students accept and adhere to treatments offered through apps for various disorders such as stress, anxiety, depression, and risky behaviors. The cost of these interventions is also reduced in light of the challenges associated with insufficient human resources [61]. Student retention is a major concern for institutions, exacerbated when there are disparities among students, contextual factors such as a pandemic, or when the field of study is more rigorous. Psychosocial factors and mental health have been shown to be critical to the academic success of college students. Studies have shown that academic anxiety has a direct impact on academic performance, which ultimately leads to attrition. The provision of mental health counseling and the willingness of students to seek support from the centers that provide these services have shown success in terms of student retention or GPA [62]. However, the complexity and resources required often exceed a university's budget and resources [63]. Academic demands will continue to place pressures on students that, by their nature, cannot be eliminated, but it is possible to reduce attrition rates by providing students with the resources, competencies, and skills to meet these challenges. These resources can become a preventative process that allows students to manage their academic challenges as part of their formal for-credit educational program. Offering courses that students can take on a voluntary basis can be a helpful strategy, but most likely students will prefer to take courses that are required and for credit and will not participate in voluntary programs regardless of their needs. Emotional support is relevant to school performance and college students' well-being. Although the causes of the systematic decline in emotional support across semesters are

unknown and could be related to the death of a family member, difficulty socializing with friends or family during the pandemic, or subjective perception, strategies for emotionally supporting students are warranted. The results of this study suggest that it is important for universities to intensify their measurement of indicators of anxiety and emotional support and to develop specific strategies to improve the mental health and wellness of students.

Limitations of this study include that all data were cross-sectional and there was no follow-up. The study involves only one University, and the data were not normally distributed, even after transformation. All students voluntarily responded to the surveys, which could suggest that those who responded felt more anxious than those who did not respond, although the scales were embedded with other survey items not specifically described as being related to anxiety or emotional support. Also, the database did not include other information that could allow discrimination among groups more broadly, such as being first-generation students or personal losses during covid, which has been shown to increase anxiety levels [43, 64]. The groups of participants included for each semester are different and the collection during the last semester dropped to the lowest level since the collection started even though the procedures to invite students and collection was the same. There is a possibility that other factors that were not considered in this study could contribute to changes in the anxiety and emotional support levels in an individual level.

## Author Contributions

**Conceptualization:** Flor Rocío Ramírez-Martínez, Maria Theresa Villanos, Marie Leiner.

**Data curation:** Flor Rocío Ramírez-Martínez, Marie Leiner.

**Formal analysis:** Marie Leiner.

**Investigation:** Maria Theresa Villanos, Sonam Sharma.

**Project administration:** Flor Rocío Ramírez-Martínez.

**Resources:** Flor Rocío Ramírez-Martínez, Marie Leiner.

**Validation:** Marie Leiner.

**Writing – original draft:** Flor Rocío Ramírez-Martínez, Maria Theresa Villanos, Sonam Sharma, Marie Leiner.

**Writing – review & editing:** Flor Rocío Ramírez-Martínez, Maria Theresa Villanos, Sonam Sharma, Marie Leiner.

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
