## [Decision Letter · Decision Letter 0]

30 Jun 2023

PONE-D-23-11873Anxiety and emotional support among first-year college students during the transition from distance to face-to-face learning during COVID-19PLOS ONE

Dear Dr. Marie,

Thank you for submitting your manuscript to PLOS ONE. After careful consideration, we feel that it has merit but does not fully meet PLOS ONE’s publication criteria as it currently stands. Therefore, we invite you to submit a revised version of the manuscript that addresses the points raised during the review process.

 Please respond to all the reviewers' comments carefully and highlight all the changes in the revised version. 

We look forward to receiving your revised manuscript.

Kind regards,

Syed Far Abid Hossain, PhD

Academic Editor

PLOS ONE

Journal Requirements:

Reviewers' comments:

Reviewer's Responses to Questions

**Comments to the Author**

1. Is the manuscript technically sound, and do the data support the conclusions?

Reviewer #1: Partly

Reviewer #2: No

2. Has the statistical analysis been performed appropriately and rigorously? 

Reviewer #1: Yes

Reviewer #2: I Don't Know

3. Have the authors made all data underlying the findings in their manuscript fully available?

Reviewer #1: Yes

Reviewer #2: No

4. Is the manuscript presented in an intelligible fashion and written in standard English?

Reviewer #1: Yes

Reviewer #2: Yes

5. Review Comments to the Author

Reviewer #1: The current study aimed to investigate anxiety and emotional support in a large sample of first-year college students transitioning from distance education to face-to-face learning during the COVID-19 pandemic.

The paper provides a good scientific contribution to the panorama of studies investigating the impact of COVID-19 pandemic on the general population's mental health, focusing on university students and their difficulties in building a productive and satisfying academic career, exacerbated during the pandemic. This study comprehensively analyzed the well-being trajectories in an extensive sample. The topic is interesting enough and qualitatively good to be published. However, some changes and explanations to improve the quality of the manuscript for his acceptance should be done.

Abstract section

In the 'abstract section' the Authors reported: "Data from 8659 freshman students were extracted from a university database". It would be useful to better contextualize the data collection, specifying the project's scope or initiative that promoted the data collection. Furthermore, the specification of the study courses to which the evaluated students belonged (humanities, mathematics, health, etc.) would be required.

Introduction section

In the 'introduction section' the bibliography is appropriate and updated. It is known that university students reported high emotional distress, anxiety and depression, low self-esteem, concentration, and learning difficulties compared to the general population, all factors that often lead to an emotional circle with a significant impact on academic performance and social relationships (listed below). In addition, it is well known that medical students suffer an increased risk of depression compared to their peers currently enrolled in non-medical university courses, and the presence of depressive symptoms seems to occur as early as the 1st year of the student's medical education (listed below), especially in women.

The conditions of medical university students have also been a research focus in Italian surveys that found that medical students highlighted issues associated with anxiety and depression, emotional distress, low perceived quality of life, problems related to alcohol consumption, and the propensity to use substances as cognitive enhancers (listed below). In addition, the Authors do not describe, as one would expect, the social changes related to the COVID-19 outbreak impact on the academic context. Distance education (DE) has replaced traditional face-to-face teaching, requiring great flexibility from university students. Additionally, home confinement compromised the possibility of fully experiencing university life, influencing academic study (i.e., delays in activities and digital platform use). The lockdown and restrictions related to the pandemic emergency probably increased students' difficulties, especially for first-year students who were starting a phase of important change for their life and future. For that reason, it would be helpful to mention some recent international studies on the impact of COVID-19 Pandemic on university students' mental health. Some authors focused on distance education's emotional and cognitive correlates during Covid-19 confinement, investigating samples of medical and health professions students on potential predictors of psychological distress and poor academic performance (listed below). Some authors focused on psychological, emotional, and cognitive correlates of social confinement in a sample of college students, integrating qualitative and quantitative analyses to identify potential predictors of traumatic distress during COVID-19. The Authors identified the thinking style "all or nothing" as the strongest predictor of traumatic distress in a sample of university students.

At the end of the introduction, the authors could report their study hypothesis.

Materials and methods section

It is not clear who promoted the survey. Is it a University psychological support Service? The Authors should make this clear. About the form information completed by each student, other characteristics such as, for example, informed consent, clinical variables (previous psychological problems and previous contacts with mental health services), and academic variables (off-course, off-site and current) were not taken into consideration of study. Authors should explain their choice or motivation.

The tables are not exhaustive for accurate data presentation. The tables are disorganized, and the figures are unclear. The authors could explain the reason for this synthesis or choice.

The authors do not report socio-demographic data of the students' sample evaluated (i.e., the year of course attended, first-year or not, foreign or off-site students, previous psychopathological conditions...). About table 1, it contains insufficient data.

Discussion section

The limitations of the study and the conclusions section are not reported.

The authors reported the need for university counselors as a useful strategy to help students cope with suffering and emotional difficulties to limit the potential structuring of psychopathological profiles over time. The authors could further discuss this aspect concerning the effectiveness that these university services have shown up to now.

In the bibliography section, not all references contain the 'doi'. Could the authors do a check?

Suggested References

1) https://pubmed.ncbi.nlm.nih.gov/32664032/

2) https://pubmed.ncbi.nlm.nih.gov/34220610/

3) https://pubmed.ncbi.nlm.nih.gov/29636088/

4) https://pubmed.ncbi.nlm.nih.gov/27923088/

5) https://pubmed.ncbi.nlm.nih.gov/31469033/

7) https://bmcpsychology.biomedcentral.com/articles/10.1186/s40359-021-00649-9

8) https://pubmed.ncbi.nlm.nih.gov/33384623/

Reviewer #2: This paper consider how anxiety and emotional support varied for entering first-year students across semesters with face-to-face and virtual instruction using repeated cross sections from Mexico. This is an important question, but I believe the manuscript needs significant revision for impact. My key concerns are as follows:

(1) The hypotheses were not clearly laid out in the introduction. I wasn't sure what the authors were testing until the methods section and even then it was not entirely clear. What are the theories this is based on?

(2) While the introduction discusses the literature and motivates the general area of interest, I did not think it was clear about what gaps this current work would fill or how the current work would contribute to the body of knowledge

(3) More background on the context is needed to understand the findings. Do new students enter every semester and is spring semester entrance different from fall semester as it is in the US? If this survey is conducted during the first 2 weeks of an immersive program prior to experience the face-to-face or virtual instruction or hybrid that is being offered that semester, why would the type of instruction matter? What happens during the immersive 2 weeks?

(4) Sample representativeness: what was the response rate and how did it compare to the population of entrants

(5) How should we interpret findings across semesters. When anxiety and emotional support differs, is this due to different characteristics of entrants at different points in time or is it about the instruction? given that the survey is taken before the student has a chance to experience the university instruction, it seems strange to me to attribute trends as an effect of the type of instruction (there are so many reasons anxiety would be increasing and declining across cohorts during the pandemic). Maybe anxiety is just declining over time and less anxious students enter university at later periods of the pandemic? I think the connection to the type of instruction is circumstantial at best. Maybe I am missing something about the institutional environment and/or maybe you are trying to make more of a claim about the high school instruction?

(6) the description of the participants is unclear (longitudinal or repeated cross sections, entering first years, etc) until we get to the later procedures section. Can these combine to be clearer?

6. PLOS authors have the option to publish the peer review history of their article (what does this mean?). If published, this will include your full peer review and any attached files.

Reviewer #1: **Yes: **Roncone Rita

Reviewer #2: No

---

## [Author Response · Author response to Decision Letter 0]

6 Sep 2023

We have submitted responses to each one of the comments and appreciate the effort and help provided to us to make this manuscript better

---

## [Decision Letter · Decision Letter 1]

29 Nov 2023

PONE-D-23-11873R1Anxiety and emotional support among first-year college students during the transition from distance to face-to-face learning during COVID-19PLOS ONE

Dear Dr. Leiner,

Thank you for submitting your manuscript to PLOS ONE. After careful consideration, we feel that it has merit but does not fully meet PLOS ONE’s publication criteria as it currently stands. Therefore, we invite you to submit a revised version of the manuscript that addresses the points raised during the review process.

We look forward to receiving your revised manuscript.

Kind regards,

Jin Su Jeong, Ph.D.

Academic Editor

PLOS ONE

Reviewers' comments:

Reviewer's Responses to Questions

**Comments to the Author**

1. If the authors have adequately addressed your comments raised in a previous round of review and you feel that this manuscript is now acceptable for publication, you may indicate that here to bypass the “Comments to the Author” section, enter your conflict of interest statement in the “Confidential to Editor” section, and submit your "Accept" recommendation.

Reviewer #2: (No Response)

Reviewer #3: (No Response)

Reviewer #4: All comments have been addressed

2. Is the manuscript technically sound, and do the data support the conclusions?

Reviewer #2: No

Reviewer #3: Yes

Reviewer #4: Yes

3. Has the statistical analysis been performed appropriately and rigorously? 

Reviewer #2: Yes

Reviewer #3: Yes

Reviewer #4: Yes

4. Have the authors made all data underlying the findings in their manuscript fully available?

Reviewer #2: Yes

Reviewer #3: Yes

Reviewer #4: Yes

5. Is the manuscript presented in an intelligible fashion and written in standard English?

Reviewer #2: Yes

Reviewer #3: Yes

Reviewer #4: Yes

6. Review Comments to the Author

Reviewer #2: Thanks for your response to my comments. In reading the new introduction, I see a better case made for the contribution and why the design might be useful for understanding the effect of virtual instruction. I also understand now the 2 hypotheses listed.

However my main remaining concerns are as follows:

(1) Who enters may vary across cohorts due to the anxiety already produced by the pandemic or other related factors

(2) It is unclear why differences in anxiety and emotional support across cohorts of entering first-year students should be attributed to switch from virtual learning to face to face and not other factors

The current framing seems misleading to me. We can see in the data that anxiety levels were higher among entering first year students when the university returned to face to face instruction than during semesters of virtual instruction. This could be due to the switch from virtual learning (though some students may have experience face to face or virtual learning previously). It may have nothing to do with the pandemic given that we do not have a pre-pandemic comparison, i.e., anxiety may always be higher in the face to face scenario. It could be something about other things that happened in Fall 2022, for instance, if the university changed who was admitted (it’s a much larger sample?) or other changes in policies that made students more anxious. Emotional support is just declining over time and seems like it could be something not related to the switch from face-to-face to virtual, which just seems part of trend.I'd really like those distinctions to be clear in the description of the study/intro and in the discussion/limitations. The comparisons are interesting, but the different patterns related to anxiety and depression around the return to in-person could be driven by different factors among these.

Additional context was useful, but I still did not see how we can think about comparison across the different cohorts from this. I still don’t understand whether the variation across Fall and Spring semesters is comparable in terms of selection into college—there do indeed appear to be differences according to Table 1. For instance, did people put off going to college because it was not in person? Background on Covid situation during this period could also be useful for framing how to think about the timing. Part of the effect over time could be length of exposure to remote instruction prior to college. Was the country in lockdown, social distancing, what was the extent of remote education in high school?

Reviewer #3: 1. The abstract can be further improved by specifying the key findings based on the scales used.

2. Kindly include review of related research i.e Related Works in a sub section, before the methodology section.

3. The presentation of the results need to simplify and easy to read/understand, as the level of statistical understanding might differs.

Reviewer #4: This study investigates anxiety and emotional support in a very large sample of first-year college students.

The research methods and the H1, H2 etc are well defined as far as the aim of the study. There is a sufficient literature reference list and provides a good scientific evidence for the impact of COVID-19 pandemic on the general population's mental health, focusing on university students. The text is interesting, timely written and it is considerd good to be published.

7. PLOS authors have the option to publish the peer review history of their article (what does this mean?). If published, this will include your full peer review and any attached files.

Reviewer #2: No

Reviewer #3: No

Reviewer #4: No

---

## [Author Response · Author response to Decision Letter 1]

30 Dec 2023

Reviewer #2: Thanks for your response to my comments. In reading the new introduction, I see a better case made for the contribution and why the design might be useful for understanding the effect of virtual instruction. I also understand now the 2 hypotheses listed.

However my main remaining concerns are as follows:

(1) Who enters may vary across cohorts due to the anxiety already produced by the pandemic or other related factors

Response:

We agree that the “why” is unknown and that individual differences can occur in the cohorts. We are not aware of any individual factors that specifically contribute to an increase in anxiety levels for each of the students in these comparisons. To respond to this possible misleading assumption, we have included the possibility that other factors could contribute to anxiety and emotional support levels in an individual level in the limitations section. The mediating role of social support in anxiety among college students has been explored in various studies which include that perceived social support is not only directly associated with anxiety, depression, and insomnia among college students during the pandemic but also indirectly connected with these mental health variables via self-control. The fact that social support seems to decrease during the pandemic in this large sample might be an interesting finding. 

In regard to the possible differences in the cohorts’ participants during this collection the university was very concerned about the levels of anxiety and implemented a wellness program to attempt to remediate these levels. There were no observable differences in the policies that affected students as far as it was reported, the admissions policies document has not been changed in 6 years until 2023. We reported this in the limitations.

(2) It is unclear why differences in anxiety and emotional support across cohorts of entering first-year students should be attributed to switch from virtual learning to face to face and not other factors

Response 

As stated in the hypothesis, our objective was to present differences between the cohorts that occurred between semesters. In the case of anxiety, it increased during the semester they returned to face-to-face classes. To respond to this concern, we have included in the limitations the possibility that other factors could contribute to reducing or increasing the scores attenuated by the sample size.

The current framing seems misleading to me. We can see in the data that anxiety levels were higher among entering first year students when the university returned to face to face instruction than during semesters of virtual instruction. This could be due to the switch from virtual learning (though some students may have experience face to face or virtual learning previously). It may have nothing to do with the pandemic given that we do not have a pre-pandemic comparison, i.e., anxiety may always be higher in the face-to-face scenario. 

Response: 

We agree, although the framework presented does not consider the pandemic the cause of the anxiety levels or decrease in the emotional support, instead it is a global environmental factor that allowed us to observe possible differences during these semesters. Our point is that in similar conditions, considering the return to face-to-face classes, the student’s anxiety seemed to increase. We are presenting the differences observed in this population. 

It could be something about other things that happened in Fall 2022, for instance, if the university changed who was admitted (it’s a much larger sample?) or other changes in policies that made students more anxious. 

Response: 

As far as we know, the university did not change anything globally and the number of students that applied and were accepted did not change in proportion in any of the semesters considered. The larger change we observed is registered statistically during the return to face-to-face classes. We have made changes in the document to report that no major changes in the admissions policies happened between these times. The fall of 2022 was a moment of return to partial normality and face-to-face classes that seem to have an influence on the anxiety levels perceived by the students, indeed we cannot report an effect based on this alone. However, the sample included is large and the overall anxiety levels seem to vary. The title of the study is now to show variations across semesters in pandemic times, without specifically pointing to returning to class as a cause, but instead to describe the large sample variations.

Emotional support is just declining over time and seems like it could be something not related to the switch from face-to-face to virtual, which just seems part of trend. I'd really like those distinctions to be clear in the description of the study/intro and in the discussion/limitations. The comparisons are interesting, but the different patterns related to anxiety and depression around the return to in-person could be driven by different factors among these.

Response: 

Emotional support seems to follow a trend indeed and that is not related to the switch from face-to-face to virtual. We have added clarity in these distinctions in the sections included by the reviewer even changing the title to suggest that we are presenting variations, although we did not changed the hypotheses. 

Additional context was useful, but I still did not see how we can think about comparison across the different cohorts from this. I still don’t understand whether the variation across Fall and Spring semesters is comparable in terms of selection into college—there do indeed appear to be differences according to Table 1. 

Response: 

The cohorts are composed of students who responded to these scales during immersion week. The differences in table 1 are due to variations in regard to the age, civil status, area of study and living conditions that are not unusual as the response was not mandatory. For example, in the studies by Andersen et al. (2022) and Wong et al. (2022) they examined the impact of college campus closures during the pandemic on students' sense of connectedness, engagement, and mobility, providing insight into the impact of pandemic-related disruptions on students' academic experiences and preferences. These studies showed population’s samples and provided valuable insights into the multifaceted effects of the COVID-19 pandemic on college students' study preferences. They highlight the importance of understanding and addressing the multiple challenges college students are facing during this unprecedented time. 

For instance, did people put off going to college because it was not in person? Background on Covid situation during this period could also be useful for framing how to think about the timing. Part of the effect over time could be length of exposure to remote instruction prior to college. Was the country in lockdown, social distancing, what was the extent of remote education in high school?

Response:

We have added background information to highlight the situation in Mexico including lockdown and social distancing. Remote learning was extensive at the high school and college level from March 2020 to August 2022. 

Reviewer #3: 1. The abstract can be further improved by specifying the key findings based on the scales used.

Response:

We have added key findings to the abstract in regard to the scales used. 

2. Kindly include review of related research i.e Related Works in a sub section, before the methodology section.

We have added the subsection requested.

3. The presentation of the results need to simplify and easy to read/understand, as the level of statistical understanding might differs.

We have made reviews to the results to simplify them and make them easy to understand 

Reviewer #4: This study investigates anxiety and emotional support in a very large sample of first-year college students.

The research methods and the H1, H2 etc are well defined as far as the aim of the study. There is a sufficient literature reference list and provides a good scientific evidence for the impact of COVID-19 pandemic on the general population's mental health, focusing on university students. The text is interesting, timely written and it is considered good to be published.

---

## [Decision Letter · Decision Letter 2]

29 Jan 2024

Variations in Anxiety and Emotional Support among first-Year college students across different learning modes (Distance and Face-to-Face) during COVID-19

PONE-D-23-11873R2

Dear Dr. Leiner,

We’re pleased to inform you that your manuscript has been judged scientifically suitable for publication and will be formally accepted for publication once it meets all outstanding technical requirements.

Kind regards,

Jin Su Jeong, Ph.D.

Academic Editor

PLOS ONE

Additional Editor Comments (optional):

Reviewers' comments:

Reviewer's Responses to Questions

**Comments to the Author**

1. If the authors have adequately addressed your comments raised in a previous round of review and you feel that this manuscript is now acceptable for publication, you may indicate that here to bypass the “Comments to the Author” section, enter your conflict of interest statement in the “Confidential to Editor” section, and submit your "Accept" recommendation.

Reviewer #2: All comments have been addressed

Reviewer #3: All comments have been addressed

2. Is the manuscript technically sound, and do the data support the conclusions?

Reviewer #2: Yes

Reviewer #3: Yes

3. Has the statistical analysis been performed appropriately and rigorously? 

Reviewer #2: Yes

Reviewer #3: Yes

4. Have the authors made all data underlying the findings in their manuscript fully available?

Reviewer #2: Yes

Reviewer #3: Yes

5. Is the manuscript presented in an intelligible fashion and written in standard English?

Reviewer #2: Yes

Reviewer #3: Yes

6. Review Comments to the Author

Reviewer #2: Thank you for your responses to my comments and clarifications. I have no additional concerns to share.

Reviewer #3: The authors have made the acceptable improvements as being highlighted during the first review and acceptable to publish this paper.

7. PLOS authors have the option to publish the peer review history of their article (what does this mean?). If published, this will include your full peer review and any attached files.

Reviewer #2: No

Reviewer #3: No

---

## [Editor Report · Acceptance letter]

26 Feb 2024

PONE-D-23-11873R2 

PLOS ONE

Dear Dr. Leiner, 

I'm pleased to inform you that your manuscript has been deemed suitable for publication in PLOS ONE. Congratulations! Your manuscript is now being handed over to our production team.

Kind regards, 

on behalf of

Dr. Jin Su Jeong 

Academic Editor

PLOS ONE